# Endometrial Heparin-Binding Epidermal Growth Factor Gene Expression and Hormone Level Changes in Implantation Window of Obese Women with Polycystic Ovarian Syndrome

**DOI:** 10.3390/biomedicines11020276

**Published:** 2023-01-19

**Authors:** Zulazmi Sutaji, Muhammad Azrai Abu, Nurainie Sayutti, Marjanu Hikmah Elias, Mohd Faizal Ahmad, Abdul Ghani Nur Azurah, Kah Teik Chew, Abdul Kadir Abdul Karim, Nor Haslinda Abd Aziz, Mohd Helmy Mokhtar, Reena Rahayu Md Zin, Zeti Azura Mohamed Hussein

**Affiliations:** 1Department of Obstetrics & Gynecology, Faculty of Medicine, Universiti Kebangsaan Malaysia, Bandar Tun Razak, Cheras, Kuala Lumpur 56000, Malaysia; 2Faculty of Medicine & Health Sciences, Universiti Sains Islam Malaysia, Persiaran Ilmu, Bandar Baru Nilai 71800, Malaysia; 3Advance Reproductive Centre, Faculty of Medicine, Universiti Kebangsaan Malaysia, Bandar Tun Razak, Cheras, Kuala Lumpur 56000, Malaysia; 4Department of Physiology, Faculty of Medicine, Universiti Kebangsaan Malaysia, Bandar Tun Razak, Cheras, Kuala Lumpur 56000, Malaysia; 5Department of Pathology, Faculty of Medicine, Universiti Kebangsaan Malaysia, Bandar Tun Razak, Cheras, Kuala Lumpur 56000, Malaysia; 6The Institute of Systems Biology, Universiti Kebangsaan Malaysia UKM, Bangi 43600, Malaysia

**Keywords:** gene expression, heparin-binding epidermal growth factor, hormonal assay, implantation window, obese, polycystic ovarian syndrome, progesterone therapy

## Abstract

Introduction: Polycystic ovarian syndrome (PCOS) is a common endocrine disorder amongst reproductive-age women, and 61% to 76% of women with PCOS are obese. Obese women with PCOS are usually burdened with infertility problems due to implantation failure. Thus, progesterone treatment is usually used to improve implantation rates. Although Hb-EGF expression is actively involved in endometrial receptivity and implantation, the data on heparin-binding epidermal growth factor (*Hb-EGF)* expression following progesterone therapy in obese women with PCOS are still lacking. Objective: To investigate the changes in serum follicle-stimulating hormone (FSH), luteinising hormone (LH), dehydroepiandrosterone sulphate (DHEA), progesterone and oestradiol levels and *Hb-EGF* expression in obese women with PCOS during the implantation window following progesterone therapy. Method: A total of 40 participants aged 18–40 years old were recruited following the provision of written consent. The participants were divided into the obese PCOS, normal-weight PCOS, obese fertile and normal-weight fertile groups. First blood collection was done before ovulation. Then, daily oral micronised progesterone (Utrogestan 200 mg) was given to the PCOS group for 10 days. The treatment was followed by a second blood collection and endometrial tissue sampling by using a Pipelle de Cornier catheter. In the fertile group, ovulation was confirmed by using ultrasound, and a second blood sample was collected on days 7 to 9 postovulation. The serum levels of FSH, LH, DHEA, progesterone and oestradiol were measured in all participants. Wilcoxon signed-rank test was used to compare FSH, LH, DHEA, progesterone and oestradiol levels during pre- and postovulation. Mann–Whitney test was performed to compare FSH, LH, DHEA, progesterone and oestradiol levels between two groups: (1) the PCOS group and the fertile group, (2) the obese PCOS group and the non-obese PCOS group and (3) the obese group and the non-obese fertile group. Result: Serum FSH levels were lower in obese women in their follicular phase than in women with normal weight regardless of their PCOS status, whereas serum LH/FSH ratios and DHEA levels were higher in women with PCOS than in women without PCOS. However, endometrial *Hb-EGF* expression was lower in the obese PCOS group than in the normal-weight PCOS group. Conclusions: Different patterns of hormonal levels and *Hb-EGF* expression levels were seen between the studied groups. However, further in vitro and in vivo studies are needed to investigate the mechanism underlying the changes in FSH, LH/FSH ratio, DHEA and *Hb-EGF* expression in PCOS after progesterone treatment.

## 1. Introduction

Polycystic ovarian syndrome (PCOS) is a common endocrine disorder that affects between 6% to 15% of reproductive-age women [1]. PCOS is characterised by a combination of signs and symptoms of hyperandrogenism, menstrual irregularity and evidence of ovarian cysts on sonography [2]. In PCOS, subfertility is an important concern. Thus, women with PCOS are the most frequent attendees of fertility clinics. However, a high failure rate following fertility treatment is observed, especially in women with PCOS and obesity. Obesity has been reported to be related to PCOS, given that 61% to 76% of women with PCOS are obese [2]. 

The low pregnancy rate of women with PCOS and obesity following infertility treatment is usually due to implantation failure. Obesity causes a considerable alteration in endometrial receptivity that affects proper embryonic implantation, thus resulting in increased pregnancy failure [3,4,5]. Implantation failure has been reported to occur in 30% to 50% of all conceptions and is strongly influenced by body mass index (BMI) [6]. 

Thus, progesterone therapy is widely used as an attempt to improve implantation failure in PCOS women. Progesterone therapy has been demonstrated to be an effective oral alternative for preventing premature luteinising hormone (LH) surges in assisted reproductive techniques [7]. It also suppresses the expression of androgen receptors in the endometrium and contributes to the improvement in PCOS-related endometrial dysfunction during the implantation window [8]. 

Implantation is the process in which the embryo attaches to the endometrial lining of the uterus before it invades the epithelium and maternal circulation to form the placenta [9]. It usually occurs within a window between days 16 to 22 of a normal 28 day menstrual cycle, 5 to 10 days after the LH surge [9]. Thus, studying the endometrium biomarkers that are expressed during the implantation window is essential for predicting successful implantation. 

HB-EGF is used as a biomarker to predict endometrial receptivity and successful embryo implantation during infertility treatment. HB-EGF is a member of the epidermal growth factor family that plays a role in follicular development, ovulation, embryo implantation and early development [10,11]. HB-EGF is mainly expressed in the luminal and glandular epithelia of the endometrium. HB-EGF expression peaks in the mid-secretive stage or on day 20 of the menstrual cycle [11]. The expression level of HB-EGF is positively correlated with endometrial thickness by acting as a mitogenic factor for human endometrial stromal cells [12]. However, data on the level of Hb-EGF gene expression in women with PCOS, especially those who are obese, remain unavailable. Thus, this study aims to investigate the difference between endometrial Hb-EGF expression and hormonal changes in circulating blood in women with obesity and PCOS and women with normal BMIs and PCOS. 

## 2. Materials and Methods

### 2.1. Study Design

This prospective study was undertaken at the Medically Assisted Conception Clinic at Hospital Canselor Tuanku Muhriz (Universiti Kebangsaan, Malaysia) from January 2019 to June 2021. Ethical approval was obtained from the Research and Ethics Committee of Universiti Kebangsaan Malaysia (UKM.FPR.SPI 800-2/28/6) and funded by National Fundamental Research Grant Scheme (FRGS/1/2018/SKK08/UKM/03/2). The trial was registered on 22 November 2019 under NCT04175002 (ClinicalTrial.gov).

### 2.2. Patient Recruitment 

A total of 40 participants aged 29–41 years old were recruited following the provision of written consent. Inclusion criteria for the study group included women who were diagnosed with PCOS that fulfilled two out of three Rotterdam criteria, such as oligo- or anovulation, hyperandrogenism and polycystic ovary on ultrasound. Normal level of thyroid function test and serum prolactin level were identified to rule out thyroid disorder and hyperprolactinemia. The inclusion criteria for the control group included healthy volunteers with confirmed fertility with at least one child, a normal level of basic reproductive hormones and a regular menstrual cycle interval. Exclusion criteria included smoking; hormonal treatment for anovulation for at least 3 months prior to sample collection; pregnancy or lactation during the previous 12 months; other systemic diseases, including endocrine and eating disorders and uterine or ovarian diseases; on any regular medication, such as hormones, herbal substance, statins or corticoids for at least 3 months before sample collection; and history of intrauterine device replacement. Clinical indices, including height, weight and BMI, were measured for each participant. The participants were divided into four groups in accordance with their PCOS diagnosis and BMI. The groups are (1) anovulatory women with PCOS and BMI greater than 27 kg/m^2^ (*n* = 10), (2) anovulatory women with PCOS and BMI lower than 27 kg/m^2^ (*n* = 10), (3) healthy fertile women with BMI greater than 27 kg/m^2^ (*n* = 10) and (4) healthy fertile women with BMI lower than 27 kg/m^2^ (*n* = 10).

### 2.3. Sample Collection and Hormonal Level Measurement

Peripheral blood sample collection was done twice for each participant and the first blood collection was done before ovulation. In the PCOS group, first blood collection was done before daily oral micronised progesterone (Utrogestan 200 mg) was given for 10 days, and a second blood collection was done on the 10th day (mid-secretory phase). For the control group, ovulation was confirmed by using a urine ovulation test, and a blood sample was collected on days 7 to 9 post-ovulation (mid-secretory phase). A total of 5 mL of peripheral blood was withdrawn from the participants for hormonal assays during each blood collection session. The serum levels of follicle-stimulating hormone (FSH), LH, dehydroepiandrosterone (DHEA), progesterone and oestradiol in all participants were measured using Cobas e 411 Analyzer (Roche Diagnostics, Basel, Switzerland) via Electrochemiluminescence technology. Endometrial tissue samples were collected from all of the participants by using a Pipelle de Cornier catheter during the second blood collection. 

### 2.4. Hb-EGF Gene Expression 

Total RNA was isolated from endometrial tissue with a RNeasy Plus Mini Kit (Qiagen) by following the manufacturer’s instructions. RNA was eluted with 30 µL of RNAase–DNAse free water, then its concentration and quality were measured. The total RNA was then reversed transcribed into cDNA by using iScript™ cDNA Synthesis Kit (BioRad, Hercules, CA, USA). SsoAdvanced™ Universal SYBR^®^ Green Supermix (BioRad) was utilised in the qPCR master mix, and the thermal cycler applied was an iCycler iQ™ Real-Time PCR Detection System (BioRad). The cycling conditions included denaturation at 98 °C for 30 s, 40 cycles of 95 °C for 15 s and 60 °C for 30 s. The melting curve was set at 65 °C to 95 °C for 5 s with an increment of 0.5 °C. Relative quantification was performed via normalisation against two housekeeping genes (GAPDH and ACTB). The stability of each housekeeping gene was assessed by using Normfinder and Genorm. Hb-EGF gene expression levels were calculated through relative quantification via the delta–delta Ct method.

### 2.5. Statistical Analysis

All statistical analyses were performed by using Statistical Package for Social Sciences for Windows 22.0. Wilcoxon signed-rank test was used to compare FSH, LH, DHEA, progesterone and oestradiol levels during pre- and post-ovulation. A *p*-value of 0.05 was considered statistically significant. Mann–Whitney test was used to compare Hb-EGF gene expression, FSH, LH, DHEA, progesterone and oestradiol levels between two groups; specifically, between (1) the PCOS group and the normal fertile group, (2) the obese PCOS group and the non-obese PCOS group and (3) the obese group and the non-obese fertile group.

## 3. Results 

### 3.1. Clinical Assessment and Categorisation 

A total of 40 women aged 29–41 years old were recruited in this study. The participants were categorised into four groups in accordance with their PCOS diagnosis and BMI. Each group consisted of 10 participants. Table 1 shows the demographic profile of all the participants. 

### 3.2. Changes in Hormone Levels

The levels of FSH, LH, DHEA, progesterone and oestradiol were measured and compared between the samples collected from the PCOS groups during preovulation (follicular phase) and after progesterone therapy or from the control groups during the mid-secretory phase (post-ovulation). Table 2 shows the mean and standard deviation of the hormones in the samples collected during pre- and post-ovulation in accordance with the study groups. In the normal-weight control group, FSH level was significantly higher during the follicular phase than in the midluteal phase (*p* = 0.028). In all groups, no other significant changes in hormone levels were identified between pre- and post-ovulation.

Hormone levels were compared between (1) the PCOS group and the control group, (2) the obese PCOS group and the normal-weight PCOS group and (3) the obese control and the normal-weight control group (Table 3). Significant differences in DHEA (*p* = 0.020) levels between the PCOS group in pre-progesterone therapy and the control group in pre-ovulation were observed. The PCOS group showed higher DHEA levels (mean rank = 18.5) than the control group (mean rank = 10.86). No significant difference in all hormone levels during pre- and post-ovulation was observed between the obese PCOS group and the normal-weight PCOS group.

Significant differences were observed in FSH (*p* = 0.013), LH (*p* = 0.017) and DHEA (*p* = 0.009) levels between the obese control group pre-progesterone therapy and the normal-weight control group during pre-ovulation. The FSH level (mean rank = 13.80) and LH level (mean rank = 13.65) of the normal-weight control group were higher than the FSH level (mean rank = 7.20) and the LH level (mean rank = 7.35) of the obese control group. By contrast, a lower DHEA level (mean rank = 6.55) was observed in the normal-weight control group during pre-ovulation than in the obese group (mean rank = 13.19).

### 3.3. Hb-EGF Gene Expression

Hb-EGF gene expression was measured and compared between study groups. A significant decrease in Hb-EGF expression (−54.63 fold change) was observed in the obese PCOS group when compared to the normal-weight PCOS group (*p* = 0.028). The means and standard deviations of Hb-EGF expression in each group are listed in Table 4. Simple linear regression analysis revealed no significant correlation between Hb-EGF expression and FSH, LH, oestradiol, progesterone and DHEA levels. 

## 4. Discussion 

Elucidating the hormonal level changes and endometrial Hb-EGF gene expression during the implantation window is crucial for improving endometrium receptivity in obese women with PCOS. In the present study, FSH level was higher during preovulation than during post-ovulation in the normal-weight fertile group (*p* = 0.028) and the normal-weight PCOS group (*p* = 0.180). However, the FSH level was lower during preovulation than during post-ovulation in the obese fertile group and the obese PCOS group (*p* = 0.153 and 0.715). Even though some of the results were not significant, the pattern of the changes in FSH levels between pre- and post-ovulation showed that women with normal weight had higher preovulation FSH levels and obese women had lower preovulation FSH levels than other women, regardless of their PCOS status. 

The increase in LH release due to the abnormal feedback mechanism of oestrogen contributes to the increase in the LH/FSH ratio in women with PCOS. Consequently, the insufficient levels of FSH leads to impaired follicular development, and increased levels of LH leads to increased ovarian androgen production [13]. A previous study found that BMI is not correlated with increased LH/FSH ratio [14]. However, the time of sample collection was not mentioned in this study. Given the changes in hormone levels throughout the menstrual cycle, standardising the time of sample collection is very crucial. Another report showed that during the second and third days of the menstrual period (follicular phase), almost 71% of women with PCOS showed elevated LH/FSH ratios of more than two [15]. However, in the present study, the mean LH/FSH ratio during the follicular phase and midluteal phase was lower than two in all of the PCOS and fertile groups. Nonetheless, during the follicular phase, the LH/FSH ratio was higher than one in women with obesity and PCOS (1.41) and in women with normal weight and PCOS (1.01), whereas the LH/FSH ratio was less than one in non-PCOS samples. Our finding is consistent with the result reported by Le et al. (2019), who suggested using a LH/FSH ratio of 1.33 during the follicular phase as the optimal cut-off value for distinguishing between PCOS and non-PCOS [16]. 

No cyclical variations in DHEA levels across the menstrual cycle have been previously reported [17]. However, overweight women had lower DHEA levels than women with normal weight [17]. High DHEA levels can be seen in 22% to 52% of women with PCOS [18]. In the present study, the DHEA level in the PCOS group was significantly higher before progesterone treatment (during the follicular phase) than in the fertile group (*p* = 0.020). Another study reported that the serum DHEA level is higher in the PCOS group during the follicular phase [19]. No significant difference in DHEA levels between the PCOS group and the fertile group was observed during the midluteal phase. However, we are unable to conclude that progesterone therapy suppressed DHEA production in the PCOS groups given that the DHEA level showed no reductions after progesterone treatment. 

High Hb-EGF expression during the midluteal phase has been reported in women with successful implantation [11] and is suggested as an implantation biomarker. In the present study, endometrial Hb-EGF expression was significantly higher in the normal-weight PCOS group (mean = 55.51, SD = 125.82) than in the obese PCOS group (mean = 0.88, SD = 0.92) with *p* = 0.028. The low expression of the Hb-EGF gene in obese women with PCOS implies a reduction in the endometrial receptivity of these women. Thus, obese PCOS women are suggested to have higher rates of unsuccessful implantation than women with PCOS and normal weight on the basis of their Hb-EGF expression levels. 

Statistical analysis showed that hormone and Hb-EGF expression levels were not significantly different even though their means were different. A large sample size could be beneficial for clarifying many nonsignificant results. The patterns of serum FSH levels, LH/FSH ratios, DHEA levels and endometrial Hb-EGF expression were found to be altered in women with PCOS, especially those with obesity. Identifying the molecular mechanism underlying these changes is crucial. Thus, further in vitro and in vivo studies should be done to identify the mechanisms involved in the pathogenesis of PCOS and progesterone treatment, especially in the case of obesity, by focusing on the effects of FSH, LH/FSH and DHEA levels and Hb-EGF expression.

## 5. Conclusions

Different patterns of hormonal levels and Hb-EGF expression between the studied groups were seen. Serum FSH level was lower in obese women during the follicular phase than in women with normal weight regardless of their PCOS status, whereas serum LH/FSH ratios and DHEA levels were higher in women with PCOS than in women without PCOS. Endometrial Hb-EGF expression was lower in the obese PCOS group than in the normal-weight PCOS group. Thus, the Hb-EGF expression level could be used as a biomarker to predict successful embryo implantation, especially among obese PCOS women. Furthermore, decisions on further treatment such as metformin and appetite suppressant to manage obesity among PCOS women can be made based on the Hb-EGF expression levels. 

## Figures and Tables

**Table 1 biomedicines-11-00276-t001:** The demographic profile of participants according to the study group.

Group Category	Mean (SD)		Race			Parity	
	Age	BMI	Malay	Chinese	Indian	Nulliparous	> Para 1
Normal weight PCOS (*n* = 10)	35.80 (2.90)	23.05 (1.46)	8	0	2	10	0
Obese PCOS (*n* = 10)	37.00 (2.83)	31.25 (1.62)	7	2	1	10	0
Normal weight control (*n* = 10)	37.10 (2.77)	24.40 (1.94)	7	2	1	0	10
Obese control (*n* = 10)	33.20 (2.74)	32.46 (1.53)	8	2	0	0	10

**Table 2 biomedicines-11-00276-t002:** Hormones level during pre- and post-ovulation or progesterone therapy according to the study group.

Group	Hormone	Mean (SD)		^a^*p*-Value
		Pre-Ovulation in Control/ Pre-Progesterone Therapy in PCOS	Post-Ovulation in Control/ Post-Progesterone Therapy in PCOS	
Normal-weight PCOS	FSH (IU/L)	6.35 (0.21)	3.80 (2.20)	0.180
	LH (IU/L)	6.35 (3.61)	2.60 (1.78)	0.109
	LH/FSH	1.01 (0.60)	0.72 (0.27)	0.109
	Estradiol (pg/mL)	153.50 (75.66)	349.5 (354.22)	-
	Progesterone (ng/mL)	0.55 (0.21)	22.95 (23.94)	0.109
	DHEA (µmol/L)	6.55 (3.32)	6.65 (1.35)	0.180
Obese PCOS	FSH (IU/L)	3.13 (1.83)	5.38 (2.06)	0.715
	LH (IU/L)	3.68 (1.78)	4.26 (1.82)	0.593
	LH/FSH	1.41 (0.76)	0.81 (0.24)	0.715
	Estradiol (pg/mL)	290.25 (228.72)	185.40 (130.49)	0.593
	Progesterone (ng/mL)	9.25 (10.66)	6.94 (8.94)	0.317
	DHEA (µmol/L)	6.10 (3.26)	5.06 (1.85)	-
Normal-weight control	FSH (IU/L)	8.53 (3.32)	5.38 (1.91)	0.028 *
	LH (IU/L)	6.15 (2.83)	5.10 (1.94)	0.575
	LH/FSH	0.74 (0.26)	0.95 (0.18)	0.169
	Estradiol (pg/mL)	214.10 (158.12)	325.56 (245.01)	0.203
	Progesterone (ng/mL)	3.16 (4.17)	2.73 (3.16)	0.878
	DHEA (µmol/L)	1.41 (1.69)	4.31 (3.11)	0.110
Obese control	FSH (IU/L)	4.15 (3.21)	5.42 (3.14)	0.153
	LH (IU/L)	3.10 (1.32)	4.51 (3.40)	0.141
	LH/FSH	0.97 (0.68)	0.88 (0.34)	0.799
	Estradiol (pg/mL)	276.63 (90.35)	212.80 (173.76)	0.241
	Progesterone (ng/mL)	2.59 (2.32)	5.47 (6.95)	0.241
	DHEA (µmol/L)	4.66 (3.29)	4.40 (3.70)	0.575

******p*-value < 0.05, ^a^ Wilcoxon signed-rank test.

**Table 3 biomedicines-11-00276-t003:** Comparison of hormones level between study groups.

Group	Hormone	Pre-Ovulation in Control/ Pre-Progesterone Therapy in PCOS		Post-Ovulation in Control/ Post-Progesterone Therapy in PCOS	
		U	^a^*p*-Value	U	^a^*p*-Value
PCOS vs. Control	FSH	140.5	0.535	108.0	0.640
	LH	152.5	0.811	84.0	0.161
	LH/FSH	141.5	0.556	84.5	0.167
	Estradiol	73.0	0.423	87.5	0.353
	Progesterone	125.5	0.411	68.5	0.165
	DHEA	24.5	0.020 *	54.0	0.121
Obese PCOS vs. Normal-weight PCOS	FSH	18.5	0.168	9.0	0.167
	LH	27.0	0.634	11.5	0.329
	LH/FSH	30.0	0.874	15.0	0.685
	Estradiol	8.0	0.796	9.5	0.394
	Progesterone	22.0	0.535	3.5	0.680
	DHEA	5.0	0.724	3.0	0.860
Obese control vs. Normal weight control	FSH	17.0	0.013 *	42.0	0.545
	LH	18.5	0.017 *	30.0	0.130
	LH/FSH	46.0	0.762	41.0	0.496
	Estradiol	34.0	0.226	43.0	0.597
	Progesterone	47.0	0.820	42.0	0.545
	DHEA	10.5	0.009 *	43.0	0.870

* *p*-value <0.05, ^a^ Mann–Whitney U test.

**Table 4 biomedicines-11-00276-t004:** Comparison of *Hb-EGF* expression level between study groups.

Group	Mean Fold Change (SD)	U	^a^*p*-Value
PCOS	28.21 (91.02)	194.0	0.871
Control	22.29 (79.16)		
Obese PCOS	0.88 (0.92)	21.0	0.028 *
Normal-weight PCOS	55.51 (125.82)		
Obese control	35.71 (108.19)	50.0	1.000
Normal-weight control	8.88 (11.83)		

* *p*-value <0.05, ^a^ Mann–Whitney U test.

## Data Availability

The authors acknowledge that the data presented in this study must be deposited and made publicly available in an acceptable repository, prior to publication.

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
