# Peer review of "Endometrial Heparin-Binding Epidermal Growth Factor Gene Expression and Hormone Level Changes in Implantation Window of Obese Women with Polycystic Ovarian Syndrome"

_biomedicines, 2023, doi:10.3390/biomedicines11020276_

Round 1

Reviewer 1 Report

The authors focus on one of the most important gynecological diseases, which is polycystic ovarian syndrome (PCOS).

Unfortunately, the study methodology seems debatable and not fully thought out. The results of hormones levels and Hb-EGF were compared in four groups of only 10 women. In "Materials and methods" the authors wrote that the age of the women was "18-40 years", while in "Results" - "29-41 years". Table 1 shows that the authors compared definitely older women than younger subjects, as the age SD ranged from 33.2 years (in obese control) to 37.1 years (normal weight control).

The results regarding to hormone levels presented in Table 2 are comparable in the pre-ovulatory and post-ovulatory periods. The only statistically significant difference is the concentration of FSH in the group of normal weight women without PCOS. All other results are identical in these periods for the women of the analysed groups. For a reviewer who is an endocrinologist and gynecologist, these results are astonishing. This is probably due to the determination of the follicular phase on different days - certainly not the results from the 3-4th day of the cycle.

Another controversial point is the diagnosis of PCOS in two groups of patients. According to the Rotterdam consensus, PCOS is defined by the presence of two of three of the following criteria: oligo‐anovulation, hyperandrogenism and polycystic ovaries, but Rotterdam criteria also require exclusion of other conditions that mimic PCOS.  Has such diagnostics been carried out correctly? Has adrenal disease been ruled out since 98% of dehydroepiandrosterone sulfate (DHEAS) is produced in the adrenal glands and not in the ovaries?

For what purpose was endometrial sampling collected after ovulation - in potentially fertile women?

Author Response

1) In the “Material and method”: Thank you for addressing the error.

- The participant age was change to “29-41 years old”

2) Rotterdam criteria also require the exclusion of other conditions that mimic PCOS:

- We included a few tests to exclude other conditions that mimic PCOS. A sentence on the tests has been added- “Normal level of thyroid function test and serum prolactin level was identified to rule out thyroid disorder and hyperprolactinemia”.

3)Endometrial sampling was taken during the luteal phase in fertile women to determine the level of Hb-EGF in a control group which is considered as a good implantation cohort. Hence, the result is comparable with those who have implantation abnormality in PCOS and the obese group.

Reviewer 2 Report

The topic of the manuscript is of interest because of the high prevalence of pregnancy loss in PCOS  patients with the significant impact of obesity.

The main remarks and suggestions are the following:

In our opinion, the title of the manuscript could mainly mention HB-EGF gene expression and hormones in obese PCOS vs nonobese PCOS patients without the stress on the progesterone therapy.

It is unclear, how the sample size was calculated.

Regarding the diagnostic criteria, Rotterdam (2003) definitions include not only two from the three criteria of PCOS, but also the exclusion of the conditions with similar symptoms, i.e. hyperprolactinaemia, thyroid disfunction, NCAH, etc.

Methods. The authors should present what exactly methods were used for the hormone’s measurement. For PCOS studies, it is essential to assess the total testosterone using a highly efficient method like Liquid Chromatography-Tandem Mass Spectrometry (LC-MS/MS) assay and to calculate  free or bioavailable testosterone. Currently, the authors mention in the Methods section that they measured dehydroepiandrosterone sulphate but use the abbreviation DHEA instead of DHEA-S throw all the text.

Results. A brief data on socio-demography, menstrual and reproductive history, and vital signs (including mFG score) are needed in the section Clinical assessment and categorisation.

The terms “preovulation” and “postovulation” for PCOS groups (Table 2-3) look inconsistent with the anovulatory status of these women.

The section on Hb-EGF gene expression looks too brief. It would be appropriate to assess the associations between changes in gene expression and previously described hormonal characteristics.

Discussion. The discussion does not reflect recent publications on the research topic, for example:

Liu S, Hong L, Lian R, et al. Transcriptomic Analysis Reveals Endometrial Dynamics in Normoweight and Overweight/Obese Polycystic Ovary Syndrome Women. Front Genet. 2022;13:874487. Published 2022 May 13. doi:10.3389/fgene.2022.874487

Author Response

1)The suggested title was changed by removing progesterone therapy.

2) Definitions include not only two from the three criteria of PCOS, but also the exclusion of the conditions with similar symptoms.
- We have included a few tests to exclude other conditions that mimic PCOS. A sentence on the tests has added- “Normal level of thyroid function test and serum prolactin level was identified to rule out thyroid disorder and hyperprolactinemia”.

3) What exact methods were used for the hormone’s measurement?
- A sentence has been added – “The hormones were measured using Cobas e 411 Analyzer (Roche Diagnostics) via Electrochemiluminescence technology”.

4) The authors mention in the Methods section that they measured dehydroepiandrosterone sulphate but use the abbreviation DHEA instead of DHEA-S throw all the text.
- Thank you for the comments. The Methods section has been corrected. Dehydroepiandrosterone is measured instead of dehydroepiandrosterone sulphate.

5) Thank you for the suggestion. Results: A brief data on socio-demography and reproductive history has been added in the section Clinical assessment and categorization.

6) The terms “preovulation” and “postovulation” for PCOS groups (Table 2-3) look inconsistent with the anovulatory status of these women.
- The term “preovulation” and “postovulation” has been replaced by “Pre-ovulation in control/Pre-progesterone therapy in PCOS” and “Post-ovulation in control/Post-progesterone therapy in PCOS”.

7) The section on Hb-EGF gene expression looks too brief. It would be appropriate to assess the associations between changes in gene expression and previously described hormonal characteristics.
- No significant correlation between Hb-EGF expression and FSH, LH, oestradiol, progesterone and DHEA levels is reported in the result section.

8) Discussion: Thank you for the suggestion of a paper citation. However, the suggested studies are not related to our current results and discussion.

Reviewer 3 Report

In the present study, the Authors investigated maternal blood levels of FSH, LH, DHEA, progesterone and oestradiol in women with obesity and PCOS and women with normal BMIs and PCOS during pre-ovulation and post-ovulation. Moreover, they investigated the endometrial HB-EGF gene expression levels in all patients during the post-ovulatory phase. The Authors demonstrated that serum FSH was lower in obese women during the follicular phase than in women with normal weight regardless of their PCOS status, whereas serum LH/FSH ratios and DHEA levels were higher in women with PCOS than in women without PCOS. Endometrial Hb-EGF expression was lower in the obese PCOS group than in the normal-weight PCOS group.

This is a very interesting study addressing a novel issue as different hormonal patterns and endometrial Hb-EGF levels between obese and normal-weight women with and without PCOS. Thus, it is likely to be of great interest to the readers of Biomedicines.

However, there are several points that the Authors must address before publication

1.      Abstract. It is not clear the correlation between “hormone levels and Hb-EGF expression”. Maybe, you should just mentioned that Hb-EGF expression is actively involved in endometrial receptivity and implantation. Please, replace Hb-EGF with Heparin-binding epidermal growth factor-like growth factor the first time you mentioned it.

2.      The main concern is about the clinical relevance of your study. Once you identify hormoral differences and Hb-EGF expression between groups, what do you suggest? They could be use as biomarkers especially during PMA? These results could help to identify new therapies or treatments for infertile women? Based on these data, could we improve the clinical conditions of these patients (e.g. DHEAS supplementation…)? These thoughts should be part of the conclusion section that you completely missed.

3.      Please, use full name only the first time you mentioned hormones name (e.g. lines 123-124 methods section)

4.      Table 1. The Authors should add more clinical details of the study population such as AMH, PARA, AFC, ethnicity…..Morever, please add range.

5.      Please, add unit of measure in all tables.

6.      It is not clear the meaning of Table 2. It is already known and expected that there are hormonal differences between pre and post-ovulation. Please, clarify

7.      Table 3. The Authors should add hormonal levels in pre and post- ovulation of both categories (e.g. pre-ovulatory FSH in control and PCOS and p-value and post-ovulatory FSH in control and PCOS and p-value). The table should have 6 columns. Please, add error standard or standard deviation.

8.      Please, rephrase results section. For example, in line 177 the Authors should just write “A significant increase was observed…..”. Moreover, it could be very useful to add a fold changes.

9.      It could be more clear to graph Hb-EGF gene expression levels with an histogram instead of a table.

Author Response

  1. Abstract. Thank you for the suggestion. The sentence has been replaced with “Although Hb-EGF expression is actively involved in endometrial receptivity and implantation, the data on heparin-binding epidermal growth factor (Hb-EGF) expression”. And Hb-EGF has been replaced with Heparin-binding epidermal growth factor the first time mentioned.

  1. Conclusion. Thank you for the comments, “Thus, the Hb-EGF expression level could be used as a biomarker to predict successful embryo implantation, especially among obese PCOS women. Furthermore, decision on further treatment such as metformin and appetite suppressant to manage obesity among PCOS women can be deemed based on the Hb-EGF expression level.” Has been added.

  1. Method. Full name only the first time you mentioned hormones name (e.g. lines 123-124 methods section) – Abbreviation is used to replace the full name of hormones name.

  1. Table 1. The demographic profile including race and parity have been added to Table 1 and the range have been added.

  1. Please, add unit of measurement in all tables- Measurement unit have been added in all tables.

  1. Table 2. It is already known and expected that there are hormonal differences between pre and post-ovulation

– Thank you for the query, table 2 was constructed to compare the changes in hormonal level between the four groups with regards to the “Pre-ovulation in control/Pre-progesterone therapy in PCOS” and “Post-ovulation in control/Post-progesterone therapy in PCOS”.

  1. Table 3. The Authors should add hormonal levels in pre and post- ovulation of both categories(e.g. pre-ovulatory FSH in control and PCOS and p-value and post-ovulatory FSH in control and PCOS and p-value). The table should have 6 columns. Please, add error standard or standard deviation.

  - Thank you for the suggestion. The hormonal level and its standard deviation can be found in Table 2. Adding them in Table 3 will make it redundant.

  1. Please, rephrase results section. For example, in line 177 the Authors should just write “A significant increase was observed…..”. Moreover, it could be very useful to add a fold changes.

- fold changes have been added to the sentence: “A significant decrease in Hb-EGF expression (-54.63-fold change) was observed in obese PCOS group when compared to the normal-weight PCOS group (p = 0.028).”

Round 2

Reviewer 1 Report

The authors have addressed my concerns and suggestions in their revised version.